# Neutrophil Recruitment and Participation in Severe Diseases Caused by Flavivirus Infection

**DOI:** 10.3390/life11070717

**Published:** 2021-07-20

**Authors:** Marina Alves Fontoura, Rebeca Fróes Rocha, Rafael Elias Marques

**Affiliations:** 1Brazilian Biosciences National Laboratory—LNBio, Brazilian Center for Research in Energy and Materials—CNPEM, Campinas 13083-100, Brazil; marina.fontoura@lnbio.cnpem.br (M.A.F.); rebeca.rocha@lnbio.cnpem.br (R.F.R.); 2Cellular and Structural Biology Graduate Program, Institute of Biology, University of Campinas (UNICAMP), Campinas 13083-865, Brazil; 3Genetics and Molecular Biology Graduate Program, Institute of Biology, University of Campinas (UNICAMP), Campinas 13083-970, Brazil; 4Department of Integrative Structural and Computational Biology, The Scripps Research Institute, La Jolla, CA 92037, USA

**Keywords:** neutrophils, flavivirus, encephalitis, hemorrhagic fever, pregnancy

## Abstract

Neutrophils are first-line responders to infections and are recruited to target tissues through the action of chemoattractant molecules, such as chemokines. Neutrophils are crucial for the control of bacterial and fungal infections, but their role in the context of viral infections has been understudied. Flaviviruses are important human viral pathogens transmitted by arthropods. Infection with a flavivirus may result in a variety of complex disease manifestations, including hemorrhagic fever, encephalitis or congenital malformations. Our understanding of flaviviral diseases is incomplete, and so is the role of neutrophils in such diseases. Here we present a comprehensive overview on the participation of neutrophils in severe disease forms evolving from flavivirus infection, focusing on the role of chemokines and their receptors as main drivers of neutrophil function. Neutrophil activation during viral infection was shown to interfere in viral replication through effector functions, but the resulting inflammation is significant and may be detrimental to the host. For congenital infections in humans, neutrophil recruitment mediated by CXCL8 would be catastrophic. Evidence suggests that control of neutrophil recruitment to flavivirus-infected tissues may reduce immunopathology in experimental models and patients, with minimal loss to viral clearance. Further investigation on the roles of neutrophils in flaviviral infections may reveal unappreciated functions of this leukocyte population while increasing our understanding of flaviviral disease pathogenesis in its multiple forms.

## 1. Flaviviruses

Flaviviruses are a group of arthropod-borne viruses composing the Flavivirus genus (*Flaviviridae* family). Flaviviruses are positive-sense, single-stranded, enveloped RNA viruses known to cause widespread morbidity and mortality throughout the world [1]. Most flaviviral infections are asymptomatic or may present with flu-like symptoms. A small proportion of patients may develop severe forms of disease vary with the etiological agent: Dengue and Yellow fever viruses (DENV, YFV) may cause hemorrhagic fever and shock [2]; Japanese encephalitis virus (JEV), West Nile virus (WNV) and St. Louis Encephalitis virus (SLEV) may cause meningoencephalitis [3]; and Zika virus (ZIKV) may cause teratogenesis in pregnant women and Guillain–Barré syndrome in male and female adults [4]. Although severe disease manifestations following infection with hemorrhagic or encephalitogenic flaviviruses, or Zika, are considerably different, they all share a common proinflammatory component manifested in target tissues.

Most flaviviruses are neglected in terms of investment in research and development and public awareness. Between October 2011 and July 2012, a study conducted in the state of Mato Grosso, Brazil, collected sera from patients presenting with acute febrile illness. The study identified, during a large dengue outbreak, that 12 out of 15 patients positive for DENV-4 RNA were also positive for Mayaro virus (MAYV) RNA, an unrelated alphavirus [5]. Another example of silent circulation of strains happened in Italy 2008, where 5 WNV-positive patients also presented Usutu virus (USUV) neutralizing antibodies [6]. Most flaviviruses lack commercially available means for disease diagnosis, specific treatments and vaccines. This situation often results in underestimated rates of infection and high rates of morbidity and mortality in affected populations, especially during outbreaks [7,8].

## 2. Neutrophils and Their Effector Functions

Neutrophils are polymorphonuclear (PMN) leukocytes, also known as neutrophilic granulocytes, which are multifaceted effector cells of the innate immune system. Neutrophils are characterized by their capacity to phagocytose a variety of objects and are specialized in engulfing and destroying pathogens (such as bacteria and fungi) and cellular debris. Despite being a heterogeneous population, neutrophils present a typical multilobed nucleus with condensed chromatin, suggestive of reduced transcriptional activity. Neutrophils originate in the bone marrow and bear numerous granules and secretory vesicles in its cytoplasm, hence granulocytes, loaded with proteases, reactive oxygen species (ROS) and several effector molecules that are involved in neutrophilic microbicidal activity [9]. Neutrophils are capable of activation, differentiation and interaction with other leukocyte populations and non-immune cells, and express a variety of receptors at the cellular surface and may secrete cytokines and proinflammatory mediators upon stimuli. Those features allow neutrophils to lead and exert considerable influence on the course of an immune response against a pathogen [10].

Neutrophils are the first leukocytes recruited to the infected or inflamed tissue. When activated by chemotactic molecules, neutrophils are recruited to tissues and perform antimicrobial and proinflammatory functions. The coordinated recruitment of neutrophils to the site of injury or inflammation might be amplified by positive-feedback loops of chemotactic signals, leading to a focal accumulation of PMNs called neutrophil swarming [11,12]. Neutrophil recruitment is induced by different chemoattractant molecules from various sources and compositions: N-formylated peptides, C3a and C5a from complement, lipid mediators such as leukotriene B4 and chemokines [13]. Accordingly, neutrophil recruitment is mediated by different receptors recognizing formyl-peptides, complement, leukotrienes and chemokines, which are all present on the neutrophil surface in physiological conditions [14]. Although chemoattractant receptors expressed by neutrophils interact with different classes of ligands, all receptors are seven-transmembrane G protein-coupled receptors (GPCRs) triggering neutrophil chemotaxis through related mechanisms. The CXC class of chemokines presenting a glutamate-leucine-arginine motif (ELR) are major inducers of neutrophil recruitment and activation [15]. Chemokine ligands CXCL1, CXCL2, CXCL3, CXCL5 and CXCL8 (in humans) bind to CXCR1 and CXCR2 receptors in human neutrophils while their orthologues bind to CXCR2 in mice. Most chemokines without ELR motif (such as CXCL4, CXCL9-11) bind to CXCR3 receptors, while CXCL12 is the only known ligand for CXCR4 [16].

Although neutrophils are short-lived cells known to have limited transcriptional activity, they express many cell surface receptors that recognize microbial patterns and proinflammatory mediators such as: innate immunity pattern-recognition receptors (PRR), cytokine receptors, Fc-receptors, chemokine receptors and adhesion molecules [14]. When stimulated, neutrophils may express chemokines, cytokines and lipid mediators [17,18]. Detection of invading pathogens through microbial PRRs triggers an antimicrobial response in neutrophils, in which phagocytosis plays a major role in restriction and elimination of infection. Phagocytic mechanisms are most effective on opsonized pathogens, which may take place by action of complement, antibodies and lectins. Surface expression of Fc-receptors, complement receptors and other classes of binding receptors are critical for neutrophilic effector functions and the control of pathogen replication. Moreover, CCL2 and CXCL10 chemokines were reported to mediate oxidative stress-induced neutrophilic inflammation in lungs [19]. The ability of neutrophils to eliminate pathogens, once phagocytosed, is directly associated with production of ROS. The nicotinamide adenine dinucleotide phosphate (NADPH) oxidase 2 (NOX2) enzyme is responsible for ROS production in neutrophils, which are released onto internalized pathogens and result in microorganism destruction [20].

Degranulation is a well-characterized effector mechanism of neutrophils and essential for their ability to fulfill their role in innate immunity [21]. At the end of the 19th century, Paul Ehrlich described two major kinds of lysozyme-carrying granules in PMN cells, the azurophilic and the specific granules. Azurophilic granules (also known as peroxidase-positive granules) are rich in mucopolysaccharide, elastase, matrix metalloproteinases, defensins and myeloperoxidase (MPO) [22,23]. Peroxidase-negative granules are loaded with varying levels of lactoferrin, an iron-binding protein participating in neutrophil adhesion and survival [24,25], as well as gelatinase, which facilitates neutrophil infiltration into the tissue [26]. When undergoing exocytosis, granule content is released and transmembrane proteins in granules are translocated to the cell surface [27].

Neutrophils are able to release their DNA into the interstitium forming a molecular complex called Neutrophil extracellular traps (NETs) [28]. NETs are networks of DNA conjugated to antimicrobial enzymes such as myeloperoxidase (MPO), elastases and histones, which can capture and kill pathogens, including viruses [28,29,30,31]. NETosis, the process of cellular death that leads to NET formation begins with the activation of neutrophils by PRRs, followed by the production of ROS and mobilization of cytosolic calcium [28]. Release of NETs depends on the activation of the protein-arginine deiminase 4 (PAD4) enzyme, which citrullinates histones H3 and H4 causing decondensation of chromatin. DNA in the neutrophil nucleus is mixed with contents from azurophilic granules and released in the cytoplasm, followed by lysis of the plasma membrane and release of NETs to the extracellular environment. Once released, NETs have an important role in containment and elimination of bacteria, fungi and viruses, but may also cause significant tissue damage [32,33,34]. Systemic inflammation and organ damage observed in mouse models of severe sepsis are correlated with neutrophil recruitment to tissues and with release of NETs. The inhibition of NET production using a NOX2 inhibitor or its degradation with DNase prevents injuries, increasing septic mouse survival up to 90%. In accordance, the levels of NETs were increased in the serum of septic patients and the levels were correlated with the scores of organ lesions studied [29,30].

After neutrophil effector functions are carried out, neutrophil-induced inflammation enters a resolution phase. The resolution of inflammation is actively mediated by leukocytes, receptors and soluble effectors, resulting in neutrophil apoptosis and clearance of apoptotic debris by macrophages, triggering the release of proresolving lipid mediators and cytokines [35]. Phagocytosis of apoptotic neutrophils reduces IL-23 production and therefore inhibits neutrophil production/release from bone marrow, ensuring homeostatic stability of the number of neutrophils in circulation [36]. Failure of this process likely contributes to chronic inflammation while release of proresolving mediators limit acute inflammation and restore normal tissue homeostasis [37].

## 3. Participation of Neutrophils in Immune Responses against Viruses

The innate immune response against viral infections starts with viral recognition by PRRs such as those from Toll-like receptor (TLR), C-type Lectin or DExD/H box RNA helicase families. These cellular sensors are present in several cell types besides neutrophils, especially in antigen presenting cells (APC) such as dendritic cells and macrophages [38]. Activation of PRRs recognizing viral DNA or RNA leads to engagement of signaling pathways leading to activation of IRF transcriptional factors and the production of type-I interferons (IFNs) and proinflammatory cytokines. Type-I IFNs act in an autocrine and paracrine way, binding to the type-I IFN receptor IFNAR and upregulating the expression of interferon stimulated genes (ISG) that culminate in an antiviral state incompatible with viral replication and dissemination. Thus, a transient or partial inhibition of interferon responses is a prerequisite for viral replication and the establishment of infection [39,40,41]. Viruses have evolved several molecular mechanisms to evade the host’s IFN responses including the inhibition of signaling pathways, ISG function and direct antagonism of type-I IFNs. Murine neutrophils exposed to IFN-λ (a type-III IFN) can express ISGs; and IFN-λ (but not IFN-β) specifically activated a translation-independent signaling pathway that reduced ROS production and degranulation in neutrophils, which was associated to control of the inflammatory process [42].

Neutrophils are able to internalize viruses bound to antibodies or opsonized by complement, and express PRRs such as TLR7, TLR8 and MDA5 that would be activated upon binding of viral genomes, depending on genome composition and structure [41]. Neutrophils are not target cells in the majority of viral infections, being either impervious to infection or nonpermissive of viral replication [29,43]. Few exceptions include the infection of neutrophils by alphavirus and evidence that hepatitis C virus can replicate in PMN cells [44], but the contribution of this event to viremia and disease development is unknown. Results obtained from in-vivo experimental models of viral infections are biased towards inherent characteristics of both host and methods and should always be considered in data interpretation before inferences on the role of neutrophils in viral infections in humans. For example, murine neutrophils are not susceptible to many human viruses such as IAV and alphaviruses [45,46], while human neutrophils have been reported to be infected by these viruses and others such as Marburg virus, Respiratory Syncytial virus (RSV) and Epstein–Barr virus [47,48,49,50,51].

Neutrophils are the main leukocyte population recruited to the respiratory tract during respiratory virus infections. Neutrophil recruitment and activation are beneficial to hosts infected with Influenza A virus (IAV), in which neutrophils contribute to recruitment of other leukocyte populations and to viral clearance. IAV directly binds to neutrophils through sialylated glycoproteins and glycolipids present on the plasma membrane, leading to the production of H_2_O_2_ [52,53]. Neutrophil production of proinflammatory cytokines such as interleukin-1β (IL-1β) and tumor necrosis factor- α (TNF-α), along with the release of ROS and other cytotoxic molecules, are important in initial phases of disease but may cause damage and tissue necrosis if uncontrolled [54,55]. Moreover, neutrophils are often involved in containing secondary bacterial infections associated with respiratory viral infections [56].

Patients with Influenza showed increased levels of CXCL8, IFN-α, TNF-α and IL-10 in plasma and/or nasal washes [57,58,59] and complement C5a in blood [60]. Studies of IAV infection in mice show that blockade of C5a minimized neutrophil recruitment to lungs and tissue injury [61] and blockade of IL-1β and IL-6 resulted in decreased neutrophil activation and chemotaxis to lungs [62,63]. RSV, Pneumonia virus (PVM) and IAV infections promote oxidative burst in murine neutrophils [56,64,65]. Complement and FcγRs receptors present in neutrophils recognize opsonized pathogens and trigger neutrophil effector functions. For example, in RSV and IAV infections, viral particles can be opsonized by surfactant protein D, facilitating neutrophil phagocytosis. Phagocytosis promotes ROS production [66,67] and fusion of cytoplasmic granules rich in hydrolytic enzymes and NADPH oxidase with the phagosome [68], where the pathogen can be destroyed. ROS production has been shown to inhibit the mammalian target of rapamycin (mTOR) kinase, triggering an antiviral response against CMV [69,70]. Furthermore, neutrophil-derived ROS promote autophagy and cell death in cellular reservoirs, which in turn are phagocytosed by neutrophils, leading to resolution of inflammation [71].

Patients suffering from chronic viral hepatitis can develop various extra-hepatic disorders, affecting the blood, kidneys and lungs, which likely involves neutrophils. The total leukocyte and neutrophil counts in the bronchoalveolar lavage fluid of patients with chronic hepatitis C were significantly greater than those of healthy controls [72]. In a separate study with chronic hepatitis C patients, impaired neutrophil phagocytosis and mild hemolysis were observed in patients with and without cirrhosis. Direct acting antiviral therapy restored neutrophil function irrespective of severity of liver disease, suggesting that neutrophil dysfunction was related to HCV replication in the liver [73]. TREM1 (Triggering Receptor Expressed on myeloid cells 1) is a proinflammatory receptor expressed by phagocytes whose soluble form is increased in patients suffering from hepatitis B or C. Lymphocytic Choriomeningitis virus (LCMV) infection is used as a model for viral hepatitis in mice and *Trem1*-deficient mice infected with LCMV present an attenuated form of hepatitis. Neutrophils were found to express TREM1 and were recruited to the infected liver and absence of TREM1 resulted in reduced levels of CCL2 and TNF-α and less immunopathology without compromising the antiviral response [74].

Recent studies have pointed out that viral pathogen-associated molecular patterns (PAMPs) induce the release of NETs, which have been associated with protection from infection. In-vitro and in-vivo studies have both demonstrated that the sticky, web-like structure of NETs can bind and sequester virions, reducing viral load and improving survival in vivo [34]. NETosis is observed in viral infections by RSV, HIV-1, CHIKV, and SARS-CoV-2 [31,32,33,75,76] and contributes to tissue damage. DNA in NETs is critical for virus sequestration, and NETs efficacy was compromised following treatment with DNase. Pentraxin 3 (PTX3) is a soluble PRR produced by several cells, including macrophages and neutrophils. Recent studies have shown that PTX3 is produced and stored in neutrophil-specific granules and released together with NETs. The mechanisms behind the functions of PTX3 in association with NETs are unclear, as the roles of PTX3 in the neutrophilic response to viral infections [77]. Moreover, MPO and α-defensins have also been found to be associated with NETs, where they would contribute to virus inactivation [78].

## 4. Participation of Neutrophils and Neutrophil-Associated Molecules in Flaviviral Diseases

### 4.1. Hemorrhagic Fevers

Viral hemorrhagic fevers (VHF) are a group of illnesses that affect multiple organ systems and are characterized by high fever, increased vascular permeability, decreased plasma volume, coagulation abnormalities, shock and hemorrhage [79]. Four different RNA viral families are known to cause VHF including *Arenaviridae*, *Bunyaviridae*, *Filoviridae* and *Flaviviridae*. Different viruses with different invading strategies and disease mechanisms can lead to VHF. Despite those differences, the severity of these individual diseases roughly correlates with the onset of vascular pathologies [80].

YFV and DENV are two flaviviruses known to cause VHF. Both viruses are mainly transmitted by *Aedes aegypti* mosquitoes and can be maintained in a sylvatic, enzootic cycle [81]. The mosquito saliva contains molecules known to modulate host inflammation, increasing cell susceptibility to the viruses. It is suggested that mosquito bite enhancement of virus infection is not related to interferon response suppression. Instead, host increased susceptibility is closely related to neutrophil-dependent inflammation after a bite [82,83]. Following a bite, mast cell degranulation markedly increases the expression of IL-1β, IL-6 and neutrophil-attracting chemokines such as CXCL1, CXCL2, CXCL3 and CXCL5. Neutrophils are recruited to the site and express high levels of IL-1β, which orchestrates the local inflammatory environment by leading to the recruitment of myeloid cells, a primary target cell for flaviviruses [83] (Figure 1).

A common underlying mechanism in VHF is a massive production of proinflammatory cytokines. High levels of cytokines such as CXCL2, CXCL8, IL-6, TNF-α are usually found in the blood of severe yellow fever and dengue fever patients [84,85]. The role of neutrophils in severe YFV infection is unclear. However, similar to what is observed in severe DENV infections, levels of CCL2, IL-1β, IL-1Ra and TNF-α are increased [86]. Recent evidence supports the involvement of neutrophils in the pathogenesis of severe dengue fever [87,88]. Given the similarities between the inflammatory pathways triggered in severe cases of Dengue and Yellow fever, neutrophils may also play an important role in YFV pathogenesis. CXCL2, CXCL8, IL-1β and TNF-α participate either in neutrophil recruitment or activation, and more studies are needed to better characterize the relationship between neutrophils and disease severity.

By comparing the transcriptional signatures of patients with early, severe and mild dengue symptoms, Hoang et al. showed that a major difference exists in the abundance of neutrophil-associated transcripts in humans [89]. Another study has shown an increase in the levels of elastase activity and other NETs components in Dengue hemorrhagic fever patients (DHF) when compared to patients presenting with mild disease, suggesting an association between neutrophil activation with disease severity. DENV infection leads to the production of CXCL8 and TNF-α which activates neutrophils leading to an upregulation of CD66b and increased production of ROS. Interestingly, increased levels of MPO-DNA complexes, major components of NETs, were found in the serum of DHF patients when compared with levels in dengue fever patients. Further, authors showed that DENV-primed neutrophils were prone to delobulation, an early feature of NET formation [88]. Further supporting the relevance of NET formation and severe dengue manifestation, in-vivo studies have shown encapsulation of platelets expressing dengue viral antigen by the NETs. Authors proposed that neutropenia, a common clinical sign in dengue patients, may be beneficial to the host, as components of NETs may aggravate dengue immunopathogenesis [90].

Overall, neutrophil recruitment during viral inoculation by *Aedes aegypti* mosquitoes seems to play a major role in early phases of infection. Further, patients with worse dengue disease outcomes present with increased neutrophil-associated transcripts along with upregulation of granulocyte activation markers, NET formation and ROS production. Similarities between proinflammatory cytokine plasma levels in both severe Dengue and Yellow fever patients indicate that the role of neutrophils should be investigated in yellow fever.

### 4.2. Encephalitis

Encephalitis is characterized by inflammation of the brain. Encephalitis may evolve from infection with neurotropic mosquito-borne flaviviruses such as JEV, WNV, SLEV and Murray Valley encephalitis virus (MVEV) or from Tick-borne encephalitis virus (TBEV) [91]. Upon the bite by a hematophagous arthropod vector, neurotropic flaviviruses are released in the skin, interact with resident mononuclear phagocytes and propagate to the draining lymph nodes, where other leukocytes become subsequently infected. Through mechanisms that have not yet been completely elucidated, neurotropic flaviviruses invade the central nervous system (CNS), infecting and activating cells, which culminates in encephalitis [92,93]. JEV alone results in approximately 60,000 deaths every year, most of which are children [94]. WNV and SLEV are reported more frequently in the elderly and the immunocompromised. Depending on patient age, mortality can reach up to 20% with the possibility of neurological sequelae [95,96].

#### 4.2.1. SLEV

Patients suffering from SLEV present with pleocytosis in the cerebrospinal fluid (CSF), in which PMN cells are the most abundant cell population. Clearance of viruses and leukocytes from CSF are indicative of resolution of infection and disease. Autopsy analyses of human brains show that flaviviral encephalitis results in brain tissue damage including neuronal death and is usually accompanied by aseptic meningitis [97].

Neutrophils are probably recruited early in the onset of SLEV, though by the time patients manifest severe symptoms and reach intensive care, leukocyte populations recruited to the brain are mostly composed of mononuclear leukocytes, primarily T lymphocytes [95]. Thus, neutrophils are known to participate in severe neurological disease caused by SLEV, but it is unknown whether neutrophils contribute to protection or disease development. In accordance with in-vivo studies with other flaviviruses, mouse models of SLEV infection have shown that viral replication and access to the central nervous system are major factors in disease development, as absence of a functional type-I IFN response renders mice completely susceptible to SLEV infection, disease and death [98]. Excessive SLEV replication in the tissues of IFNAR^−/−^ mice result in early and aggravated onset of neurological disease, including the production of the neutrophil-chemoattractant CXCL1 and increases in MPO levels in infected brains [95,98].

As mentioned previously, adult WT mice are resistant to flavivirus infection via peripheral inoculation routes. Using a mouse model of SLEV infection that employs direct inoculation of virus in the brain of immunocompetent adult mice, researchers were able to recapitulate most features of St. Louis encephalitis seen in human infection [95]. Besides SLEV replication in the CNS, brain damage and death, researchers reported the recruitment of neutrophils to infected brains using flow cytometry. Neutrophil recruitment directly correlated with disease development, and was preceded by the expression of proinflammatory cytokines and the chemokine CXCL1, suggesting that this chemokine and the CXCR2 receptor may play a role in neutrophil recruitment to SLEV-infected brains [95] (Figure 2).

#### 4.2.2. JEV

The overall symptoms of Japanese encephalitis are similar to other encephalitic flaviviruses. Given the high degree of identity between encephalitic flaviviruses, antibody cross-protective immunity is observed. Brain damage and neuronal loss during JEV infections are associated with increased inflammation marked by increased levels of TNF-α, IL-1β, CCL5, CCL2 and NO [99,100]. JEV infection in humans and mice leads to cellular infiltration in the brain and spleen with major increases in neutrophil counts in the periphery and in the CSF. Serological examinations of patients infected with JEV have shown a worse disease outcome associated with a higher percentage of neutrophils and a lower percentage of lymphocytes [101,102].

The link between microglia and neutrophil influx to the brain is not clear. Singh et al. demonstrated how JEV-induced CCR2 expression regulates microglia activation and is associated with an increase in NO and proinflammatory cytokines TNF-α and IFN-γ [103]. Moreover, patients with worse JEV encephalitis outcomes have increased and persistent levels of CXCL8 in the CSF, suggesting neutrophils are associated with worse disease outcomes [104]. Neutrophils were shown to play an important role in early antiviral defense against JEV by degrading phagocytosed JE virions. Conversely, JEV-stimulated neutrophils release significant amounts of ROS, aggravating tissue damage and neurotoxicity [105]. More studies are needed to elucidate how neutrophils act as “double-edged swords” in JEV encephalitis.

#### 4.2.3. WNV

Neutrophils accumulate in the CSF during WNV encephalitis. Neutrophilic meningitis is seen in approximately 50% of patients presenting with WNV neuroinvasive disease [106]. As stated, the role of neutrophils in disease development is not clear. The role of polymorphonuclear cells during in-vivo WNV infection was investigated and discussed by Bai and colleagues [107], who observed that macrophages quickly express high levels of *Cxcl1* and *Cxcl2* upon infection, leading to the recruitment of polymorphonuclear cells. Depletion of polymorphonuclear cells (PMN) after WNV infection led to increased viremia and early mortality. In contrast, PMN depletion prior to infection in CXCL2-deficient mice resulted in lower viremia and increased survival [107]. This data illustrate how neutrophils can have both protective and pathogenic roles in WNV encephalitis depending on timing of neutrophil depletion.

Experiments using human and mouse cells have shown that the proinflammatory chemokine osteopontin (OPN) is increased during WNV infections. OPN-deficient mice infected with WNV show similar levels of PMNs in blood and peripheral organs in comparison to infected wild-type mice. The lack of OPN leads to decreased PMN infiltration and WNV viral load in mice brains, suggesting that OPN expression facilitates the access of both PMN and WNV to the brain. Authors proposed a mechanism of neuroinvasion for WNV in which PMNs act like “Trojan Horses”, although neutrophils are not considered target cells nor support flavivirus replication [108].

#### 4.2.4. MVEV

MVEV is a mosquito-borne flavivirus endemic to northern Australia. In-vivo studies using mouse models of infection have shown that peripheral inoculation of MVEV causes encephalitis in an age-dependent fashion [109]. As the elderly are a major risk group for encephalitic flavivirus, this MVEV mouse model is considered a useful model to study arbovirus-mediated encephalitis [110]. Neurotropism of MVEV in mice is evident and takes place 4 days after peripheral inoculation. On day 5 p.i., MVEV reaches the anterior olfactory nucleus, the piriform cortex and the hippocampus in the brain, following detection in the cerebral cortex, caudate putamen, thalamus and brain stem up to day 9 p.i. [110,111]. 

Matthews and colleagues used electron microscopy to show that lymphocytes, macrophages and especially neutrophils are recruited to infected, inflamed brains. Most leukocytes were apoptotic and shown to be abutted onto neurons [112]. Neutrophils have a major role in the development of MVEV encephalitis on day 5 p.i., and neutrophil infiltration to the CNS causes an increase in the levels of TNF-α and the neutrophil-attracting chemokine CXCL1 in the CNS. Moreover, increased levels of inducible nitric oxide synthase (iNOS) are also detected and correlates neutrophil infiltration and the onset of disease. Experiments involving the depletion of neutrophils and inhibition of iNOS led to prolonged or increased survival rates, suggesting a direct link between neutrophil recruitment to the brain, local expression of iNOS and the severity of MVEV encephalitis [110].

#### 4.2.5. TBEV

TBEV is a flavivirus exclusively present in colder regions of Europe and Asia. TBEV is transmitted to humans mainly by *Ixodes ricinus* ticks [113]. Severe disease caused by TBEV manifest as encephalitis, similar to mosquito-borne encephalitic flavivirus, as indicated the virus name [113,114]. The study by Plekhova et al. showed the presence of TBEV in the cytoplasm of primary guinea pig neutrophils, using indirect fluorescence. Authors also observed that neutrophils had an increase in chromatin condensation and that several neutrophils were apoptotic. As previously mentioned, there is no robust data indicating that flavivirus can infect neutrophils in vivo [115]. Thangamani et al. have studied the transcriptional immune profile in early stages of TBEV infection. Experiments in mice suggest a neutrophil-dominated immune response early after blood meal, inferred by the increase in transcriptional levels of neutrophil-associated chemokines and cytokines. The study points out mononuclear phagocytes and fibroblasts as the primary target cells for TBEV in the skin using immunohistochemistry to detect viral proteins. Although neutrophils were markedly increased at the tick feeding site in the skin, authors did not observe the presence of TBEV antigens in neutrophils, suggesting that neutrophils play an important role in host immunomodulation after blood meal and virus transmission, but are not infected by TBEV [116].

### 4.3. Implications in Pregnancy

In “Experimental studies of congenital malformations”, James G. Wilson compiled all known teratogenic agents at that time, defined as capable of causing abnormalities in embryonic or fetal development [117]. Teratogenic agents were classified into several groups: drugs and chemicals; physical agents (e.g., temperature); growth and metabolic inhibitors; maternal nutritional deficiencies; endocrine imbalances; and infections (known as TORCH agents nowadays). Later on, in 1977, Wilson defined the signs of abnormal development as death, malformation, growth retardation and functional disorder, which includes neurological dysfunction and effects on behavior and cognition that may manifest later in life [118]. The embryonic stage is the most sensitive period to teratogen action, including viral infection, and takes place from two to eight weeks post conception, a period when women may not be aware of pregnancy [119]. In fact, infections during pregnancy are considered to be one of the major causes of maternal, fetal and neonatal mortality and morbidity [120]. Following the events of the ZIKV epidemics in 2015, this flavivirus has been classified as a teratogen among TORCH agents. ZIKV infection led to thousands of Congenital Zika Syndrome (CZS) cases, mostly in Brazil [121]. The literature reports that not only ZIKV, but also JEV, WNV and SLEV infections during pregnancy may lead to intrauterine-growth restriction (IUGR), congenital malformations and/or fetal demise, the ultimate manifestations of the action of teratogenic agents [118].

During pregnancy, the maternal organism undergoes several adaptations to tolerate the presence of the fetus and also support its development until parturition. Those are immunological, physiological and hormonal adaptations taking place at the placenta and maternal–fetal interface [122,123], but also systemically [124]. Evidence suggests an increased susceptibility of pregnant women to viral infections due to aforementioned changes in the maternal immunological profile [125], notably a shift in towards a predominantly humoral immune response rather than cytotoxic [126]. Conversely, the number of circulatory PMN cells increases progressively throughout gestation, in association to increasing G-CSF plasma levels and peaking in the third trimester. During labor, neutrophil density is selectively greater in the lower uterine segment, presenting a rich source of inflammatory mediators such as eicosanoids, collagenase, elastase, IL-1β and TNF-α [127,128,129]. Baseline neutrophil activation changes during pregnancy, characterized by increased expression of CD11b and higher responsiveness to stimuli, such as increased phagocytosis, degranulation, NETs release and higher production of ROS [130,131].

In the female reproductive tract, PMN cells dispose of invading pathogens by phagocytosis or degranulation [132]. However, their presence and activity are tightly regulated by steroidal hormone action on the epithelial cells from the female reproductive tract. The activation of estrogen receptor ESR1 in these cells downregulates epithelial factors required to initiate transepithelial migration, impairing the recruitment of neutrophils to vaginal tissues, but ensuring their integrity during pregnancy [133]. This regulatory network prevents undesirable immune activation, which often lead to gestational disorders such as pre-eclampsia, placental insufficiency and fetal growth restriction [134]. However, recurrent infections may disrupt this balance, enhancing placental constitutive expression of the neutrophil-chemoattractant CXCL8 [135], and leading to the accumulation of invading fetal and maternal neutrophils in amniotic fluid [136]. In fact, high levels of chemokines CXCL10 and CXCL8 were found in the sera of preeclamptic mothers [137], where CXCL8 activates circulatory neutrophils and induces the release of NETs in the intervillous space of preeclamptic placenta [138]. Moreover, neutrophil abnormal activity is associated with severe pregnancy disturbances such as recurrent fetal loss, gestational diabetes mellitus, IUGR and preeclampsia [138,139].

Although flaviviruses have been identified as human viral pathogens for more than a century [140], the impacts of flavivirus infection on pregnancy outcomes are not well understood. Except for ZIKV, the literature of flaviviral infection during pregnancy is limited, often resumed on reports of transplacental infection. The mechanisms by which infection occurs and how it affects embryo or fetus development remain to be explored. For instance, DENV infection during pregnancy has been associated with increased risk of hemorrhagic fever and shock [141]. DENV infection in early pregnancy was associated with fetal loss and probable transplacental transmission [142,143], while perinatal infection may result in neonatal infection [144,145]. Two reports describe YFV congenital infection: the first, dating from 1940, described two cases of fatal maternal infection mid-gestation, in which autopsy of one patient showed retro-placental hemorrhage, generalized steatosis in the fetus, extensive bleeding and leukocyte infiltration in fetal liver, lesions in the hepatic gland and hemorrhages in the intestinal tract [146]; the second documented case describes the onset of yellow fever symptoms on a female patient three days before delivery, resulting in newborn fatal infection, in Brazil [147].

Regarding encephalitogenic flaviviruses, the natural and experimental JEV infection in swine led to fetus mummification and stillbirths [148,149]. The human transplacental infection with JEV was described for the first time in 1980, during an outbreak of Japanese encephalitis. The effects of JEV infection during pregnancy varied from parturition of apparently normal children to abortion, furthermore, it was possible to isolate JEV from brain, liver and placental tissues from an aborted fetus [150]. A mouse model of intrauterine infection with SLEV showed severe neurological outcomes in fetuses, and the severity of the disease depended on the gestational day of infection [151]. There is no data on maternal or fetal risks associated with SLEV congenital infection in humans [152]. In 2002, the CDC published the first case report on intrauterine WNV infection, the 2-day old infant presented positive serology for WNV, chorioretinal scarring and severe brain abnormalities [153,154]. An anomaly rate of 10.6% was reported in a clinical study of pregnant women infected with WNV in the USA, which is almost twice the rate of 5.5% for the general population and included four cases of miscarriage and four preterm deliveries. Of 55 infants born at term, most did not present anti-WNV IgM in cord serum, but some individuals developed a degree of malformation such as abnormal growth, aortic coarctation, glycogen storage disease type 1, cleft palate, Down syndrome, microcephaly, polydactyly or lissencephaly [155]. Moreover, WNV and Powassan virus (POWV, a tick-borne flavivirus) are capable of infecting and damaging human placental explants or pregnant mice placentas [156]. Whether these outcomes are caused by flaviviral infection *per se* or by dysregulation of immune response remains to be explored, along with possible molecular mechanisms involved.

The ZIKV epidemic in 2015 brought a significant increase in cases of microcephaly and congenital malformations in Brazil, and prompted the establishment of in-vitro and in-vivo models of infection to better understand ZIKV biology and pathogenesis [157]. Zika was shown to be a neurotropic flavivirus, capable of infecting not only neuronal cells but also trophoblasts in placenta and the developing embryo, which ultimately causes the Congenital Zika Syndrome (CZS). The most common clinical manifestations of CZS include microcephaly, ventriculomegaly, intracranial calcification, ocular abnormalities and hearing loss [158,159,160]. Pregnant women with symptomatic ZIKV infection showed higher viral load in sera than nonpregnant women [161]. In addition to other proinflammatory chemokines, Camacho-Zavala and colleagues found high plasma levels of CXCL8, which correlate directly to neutrophil activation [161] (Figure 3). Moreover, the amniotic fluid from ZIKV-positive pregnant women whose children presented microcephaly revealed exceedingly high levels of cytokines and growth factors: IL-15, CCL11, CXCL10, G-CSF, IL-10, IL-1β, TNF-α, CXCL8, CCL2 and CCL5 [162]. These data corroborate the proinflammatory profile found in pregnant Zika patients and are suggestive of neutrophil involvement, as G-CSF, CXCL8, IL-1β and TNF-α are involved in neutrophil activation and effector functions. Finally, cerebrospinal fluid from infants with ZIKV-induced microcephaly showed a maintenance of the inflammatory environment by the presence of IFN-α, CXCL10 and CXCL9 after birth [163].

Transcriptome profile of human umbilical vein endothelial cells (HUVEC) infected in vitro with ZIKV revealed upregulation of IL-15, CCL5, HGF, LIF, M-CSF, CXCL1 and CXCL12 24 h post infection. Cells infected with the Puerto Rican strain, a representative of Asian ZIKV strains, showed increased levels of IL-1β, IL-10, CCL5, G-CSF, CSF, CXCL1 and CXCL12 cytokines and reduction of IL-15, IL-16, HGH, PDGFbb and CXCL9 cytokines. Inoculation of the African strain in HUVEC cell cultures resulted in a higher expression of CCL2, CCL5, bFGF, G-CSF, LIF, M-CSF, CXCL1, and CXCL12 and a decrease in IL-15, HGH and CXCL11 [164]. Recurrent expression of CXCL1 and CXCL12 suggests that ZIKV infection of umbilical endothelial cells may lead to neutrophil recruitment to the umbilical cord.

Placentas infected with ZIKV develop a type-III IFN response in the decidua. Usually, IFN-λ has been shown to be protective against infections in mucosal surfaces, inducing antiviral responses with minimal damage to tissues. In the context of the placenta, IFN-λ is also tissue-protective, sustaining the placental role as a barrier and by counteracting infection through induction of ISGs. Human trophoblast cells isolated from full-term placentas were refractory to ZIKV infection due to IFN-λ1 expression [165]. Neutrophils express high levels of the heterodimeric IFN-λ receptor IFNLR1/IL10RB [166], resulting in a repressed state of neutrophils’ effector functions induced by type-III IFN signaling, exemplified by reduction in neutrophil infiltration, degranulation, ROS production and NETs release [167]. In the early stages of ZIKV placental infection, typical flavivirus antagonism of IFN responses result in downregulation of IFN-λ2, leading to an aberrant neutrophil response and CZS [168]. CZS is later accompanied by a strong type-I IFN response, increased expression of IFIT5 and decrease of ISG15 mRNA expression levels [169]. Studies using mouse models of ZIKV infection show that fetal and placental type-I IFN responses play a major role in promoting placental damage and fetal demise in ZIKV congenital infection [169,170]. The activation of a robust type-I IFN response may lead to tissue damage through induction of chemokine expression and recruitment of neutrophils, proinflammatory monocytes and T lymphocytes [171]. High levels of IFN-α early in pregnancy may cause angiogenic imbalance in the placenta, associated with higher risks for preeclampsia [172]. Hence, the proinflammatory environment promoted by ZIKV infection during pregnancy indicates that neutrophils may be recruited to infected tissues and that neutrophil functions may be dysregulated.

## 5. Final Considerations

Significant advances have been made towards a better understanding of severe forms of flavivirus-induced disease, notably on the identification of target cells for viral replication, characterization of proinflammatory mediators produced during infection and the roles of antiviral IFN responses in restricting viral replication and disease. Patients suffering from flaviviral diseases are given supportive care, as all flaviviral diseases still lack specific treatment. A lack of therapeutic options in flaviviral diseases are partially due to our incomplete understanding of severe disease pathogenesis, regardless of whether a mild disease evolves to encephalitis, hemorrhagic fever or congenital alterations.

Literature collected in this manuscript indicates that chemokines recruit neutrophils to sites of inflammation, where they participate in the immune response against flaviviruses and may, in some cases, contribute to immunopathogenesis and disease aggravation, as summarized in Table 1. Complex neutrophil responses take place as effector mechanisms such as NETs likely contribute to virus restriction, but chemokine-dependent neutrophil recruitment to target organs and the release of proinflammatory mediators, e.g., in the placenta or the CNS, are associated with tissue damage and disease development. In contrast to abundant studies on the role of neutrophils in bacterial infections or viral infections of the respiratory tract (e.g., Coronaviruses, Influenza viruses), the study of neutrophils in flavivirus infections is neglected. New studies focused on reverting such a limitation are likely to result in the discovery of new molecular mechanisms, and inspire the design of therapeutic strategies against disease and facilitate the development of candidate treatments.

## Figures and Tables

**Figure 1 life-11-00717-f001:**
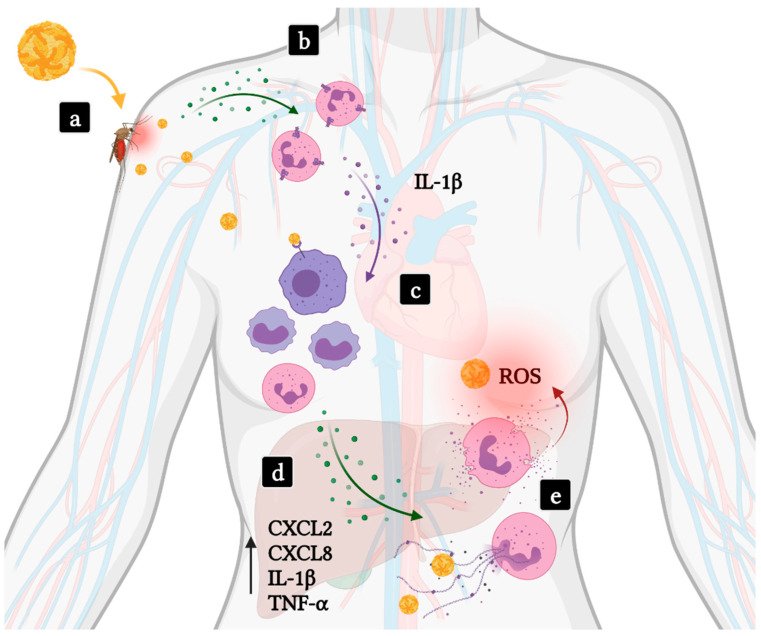
Involvement of neutrophils in hemorrhagic fever and shock induced by flavivirus infections. (**a**) Viruses such as DENV or YFV (yellow viral particles) are transmitted to human hosts through an infected mosquito bite. (**b**) Local inflammation results in the recruitment of neutrophils to the infected site. (**c**) Activated neutrophils amplify inflammation secreting IL-1β. Subsequent activation and recruitment of mononuclear phagocytic leukocytes, target cells in flavivirus infections, promotes viral replication and dissemination of infection. (**d**) Severe disease takes place as high levels of proinflammatory cytokines are produced, including those participating in neutrophil recruitment and activation, and result in systemic inflammation and multiorgan failure. (**e**) Levels of ROS and MPO-DNA complexes correlate with disease severity. Green dots—Neutrophil chemoattractants; Purple dots—IL-1β.

**Figure 2 life-11-00717-f002:**
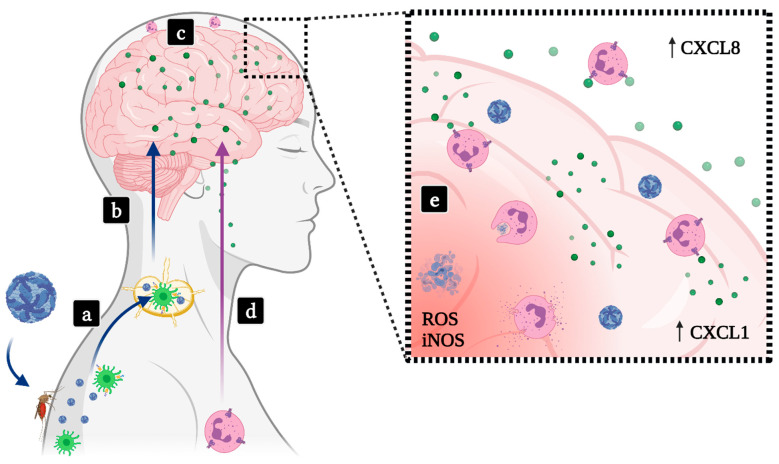
Involvement of neutrophils in flavivirus-induced encephalitis. (**a**) Encephalitic flavivirus such as WNV, JEV, SLEV, MVEV or TBEV (blue viral particles) are transmitted to vertebrate hosts through a mosquito or tick bite, reach resident mononuclear phagocytes in the skin and later disseminate to lymph nodes. (**b**) Neurotropic flaviviruses cross the blood–brain barrier through uncertain mechanisms, infecting cells in the CNS. (**c**) Infected neurons and activated glial cells release neutrophil-attracting chemokines in the CNS, initiating the development of encephalitis or meningoencephalitis. (**d**) Circulating neutrophils and other leukocytes are recruited to the CNS, resulting in CSF pleocytosis often observed in patients and consolidating the onset of severe disease. (**e**) High levels of CXC chemokines in the brain and CSF contribute to neutrophil activation and trigger neutrophil effector functions in situ, which may contribute to tissue damage, sequelae or death. Green dots—CXC chemokines.

**Figure 3 life-11-00717-f003:**
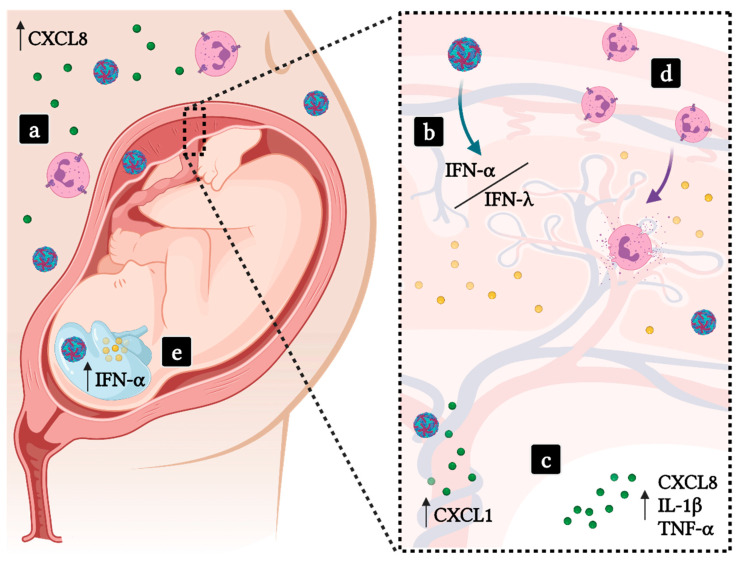
ZIKV congenital infection and the probable participation of neutrophils in CZS. (**a**) High levels of CXCL8 are found in the serum of pregnant women infected with ZIKV (blue/red viral particles). (**b**) ZIKV is able to trespass the placental barrier, the persistence of the infection leads to an imbalance on Type-I and Type-III IFN responses, which may lead to tissue damage. (**c**) Amniotic fluid from CZS pregnancies present high levels of neutrophil-chemoattractants and proinflammatory cytokines and ZIKV-infected human umbilical vein endothelial cells are sources of CXCL1. (**d**) Recruited neutrophils respond to pathogenic activation of Type-I IFN responses in patients. (**e**) ZIKV may infect fetal tissues, inducing a proinflammatory environment that promotes neuronal damage and the onset of CZS. Green dots—CXC chemokines; Yellow dots—IFNs.

**Table 1 life-11-00717-t001:** Chemokines and cytokines associated with neutrophil functions in infection and severe diseases caused by flaviviruses. For each flavivirus addressed in this review (column “Virus”), a list of chemokines and cytokines is provided in the respective column and accompanied by a brief description on how these mediators affect neutrophil function and flaviviral severe disease development (column “Involvement”). DENV—Dengue virus, YFV–Yellow Fever virus, SLEV—St. Louis Encephalitis virus, JEV–Japanese Encephalitis virus, WNV–West Nile virus, MVEV—Murray Valley Encephalitis virus, TBEV—Tick-borne Encephalitis virus, ZIKV—Zika virus.

Virus	Chemokines/Cytokines	Involvement
Flaviviruses transmitted through *Aedes* mosquito bite	CXCL1, CXCL2, CXCL3, CXCL5, IL-1β, IL-6	Promote local inflammation and recruitment of target cells to the skin.
DENV, YFV	CXCL2, CXCL8, IL-6, TNF-α CCL2, IL-1β	Promote neutrophil activation in dengue. High expression was associated with increased disease severity in yellow fever.
SLEV	CXCL1	Neutrophil recruitment to infected brain tissue.
JEV	CCL2, CCL5, TNF-α, IL-1β	High levels were associated to brain damage and neuronal loss.
	CXCL8	Accumulation in CSF in severe cases.
WNV	CXCL1, CXCL2	Involved in PMN recruitment to infected tissues.
	OPN	Facilitates infiltration of virus and PMNs in the brain.
MVEV	CXCL1, TNF-α	Neutrophil recruitment and activation, also related to increased disease severity.
TBEV	CCL2, CCL12, CXCL1, CXCL2, CXCL5, IL-6, IL-10	Upregulated in cutaneous tissue during early stages of infection.
ZIKV	CCL11, CXCL10, CXCL8, CCL2, CCL5, G-CSF, IL-1β, TNF-α	High levels in the amniotic fluid from infected mothers whose children presented microcephaly.
	CXCL10, CXCL9, IFN-α	Associated with persistent inflammation in the CSF of infants with microcephaly.
	CXCL1, CXCL12	Probably associated with neutrophil recruitment to umbilical cord during ZIKV infection.
	Type-III IFN	Protective effect against ZIKV placental infection, also downregulates neutrophils effector functions.
	Type-I IFN	Generally protective against virus infections. However, may lead to aberrant neutrophilic response in CZS and placental damage.

## Data Availability

No new data were created or analyzed in this study. Data sharing is not applicable to this article.

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
