# Peer review of "Neutrophil Recruitment and Participation in Severe Diseases Caused by Flavivirus Infection"

_life, 2021, doi:10.3390/life11070717_

Round 1
Reviewer 1 Report
The manuscript is interesting and based on extensive literature. However, the title should be slightly modified to suit the Special Issue entitled "Chemokines and Their Receptors".
Author Response
We appreciate the positive comments from reviewer 1. We changed the manuscript title to "Neutrophil recruitment and participation in severe diseases caused by Flavivirus infection" following his/her suggestion.
Reviewer 2 Report
The review article presented here is well written and describes the role neutrophils plays during flavivirus infections. I have no additional comments/suggestions for the manuscript.
The review article presented here provides a detailed collection of role of neutrophils in flavivirus infections. The review is well written with definitive sections. The figure schematics presented are well designed. I just have one comment regarding some abbreviations and way some words are presented. This is just to make the review more understandable.
- Line 100 mentions ROS but do not provide the full name for it.
- Line 145, 157 mentions type I IFN’s. I would suggest writing as type-I IFN’s. This is throughout the manuscript.
- Please expand TNFa as well.
I would suggest providing a list of secreted chemokines and cytokines associated with the different flavivirus infections. This will help to get a quick overview to the readers.
Author Response
We apreciate the positive comments from reviewer 2. Please find a point-by-point reply to the reviewer's comments below:
1 - The acronym ROS and its definitition (reactive oxygen species) is located in line 62-63 of the manuscript.
2, 3- corrections were made as requested
We also included a table (Table 1) in pages 4 and 5 listing the immune mediators associated with each infection, as suggested by the reviewer.
Reviewer 3 Report
Thank you for being selected to revise the review paper entitled: “The participation of neutrophils in severe diseases caused by Flavivirus infection”.
The presented manuscript is written in a thoughtful and understandable way. It is divided into individual sections in which flaviviruses are characterized, as well as neutrophils and their contribution to the immune antiviral response. It briefly summarises the role of neutrophils in severe diseases caused by flaviviruses. In terms of content, the information was presented fairly and accurately. What is more, clear and extremely carefully made drawings deserve special mention. My comments below are only intended to enrich the manuscript.
- In section 4.2. I propose to add two short subsections: MVEV (Murray Valley encephalitis virus) and TBEV (Tick-borne encephalitis virus), according to 4.2.1. – SLEV, 4.2.2. – JEV and 4.2.3. – WNV. That would provide a comprehensive knowledge about all cases that were mentioned in the introduction of Encephalitis (4.2.) section.
- I propose to create a short table in which the authors would present and summarise the involvement of e.g. mentioned chemokines in diseases described in the manuscript.
The authors cited and described a lot of available scientific studies. The presented manuscript is the valuable and interesting summary and I recommend this paper for publication in Life journal after minor revisions.

Author Response
We appreciate the positive comments from Reviewer 3. Please find a point-by-point reply to the reviewer's comments:
1 - Sections were included in the manuscript as suggested, in pages 8 and 9.
2 - We created such a table (Table 1) per suggestion of reviewers 2 and 3, included in pages 4 and 5.